# Enabling Tool Use of Reasoning Models Without Verifiable Reward via SFT-RL Loop

## Abstract

Large reasoning models have shown remarkable capabilities, but their internal knowledge is limited, restricting their ability to solve complex tasks. An attractive solution is to integrate external tools—such as Python for math reasoning or search engines for knowledge-intensive queries. Yet, teaching models to use tools effectively remains a significant challenge. Existing approaches often depend on reinforcement learning (RL) with accuracy-based verifiable rewards or cold-start pipelines that perform supervised fine-tuning (SFT) followed by an RL stage. These methods are shown to be notoriously unstable, prone to entropy collapse or convergence to suboptimal behaviors. The problem is compounded in real-world tool-use scenarios where accuracy signals are either unavailable or unverifiable. To address this, we propose `SR-Loop`, a general training framework that alternates between SFT and RL phases without relying on accuracy-based rewards in the RL stage. The SFT phase preserves output structure and constrains harmful exploration by imitating expert demonstrations, while the RL phase encourages discovery of new behaviors and improvements beyond the initial policy. By repeatedly cycling between these phases, `SR-Loop` achieves stable learning and progressively enhances tool-use capabilities using only structural and execution-based rewards. Experiments show that `SR-Loop` not only prevents training collapse but also delivers competitive performance on complex tool-use reasoning tasks—without requiring explicit accuracy supervision during RL. Moreover, the framework generalizes beyond tool use, proving effective for training general reasoning models even in settings without external tools.

## 1 Introduction

Large reasoning models (LRMs) (Guo et al., 2025; Yang et al., 2025; Seed et al., 2025; Team et al., 2025) have demonstrated strong capabilities in handling complex reasoning tasks (Wang et al., 2024a; Hsiao et al., 2025; Shi et al., 2025; Qu et al., 2025). Despite their impressive performance, these models are inherently constrained by the static nature of their internal knowledge acquired during training (Gao et al., 2023; Zhu et al., 2025; Wang et al., 2024b; Cheng et al., 2024; Matarazzo & Torlone, 2025). To address this limitation and further enhance their capabilities, recent research has focused on integrating external tools into the reasoning process. For instance, LRMs can be trained to generate and execute Python code for mathematical problem-solving, to query web search engines to retrieve real-time, domain-specific, up-to-date information, or to call a "worker" LLM with a dedicated role (Guo et al., 2024; Team et al., 2025; Li et al., 2025b; Jin et al., 2025; Feng et al., 2025).

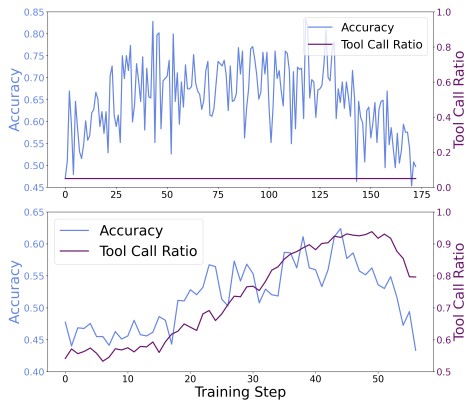

Figure 1: **(Top)** Pure RL with tool learning template and reward. **(Bottom)** Cold-start with SFT followed by RL. **Both eventually lead to performance decline.**

The dominant training paradigms for LRMs either rely on reinforcement learning (RL) with verifiable rewards or adopt a cold-start strategy that combines supervised fine-tuning (SFT) with an RL phase.

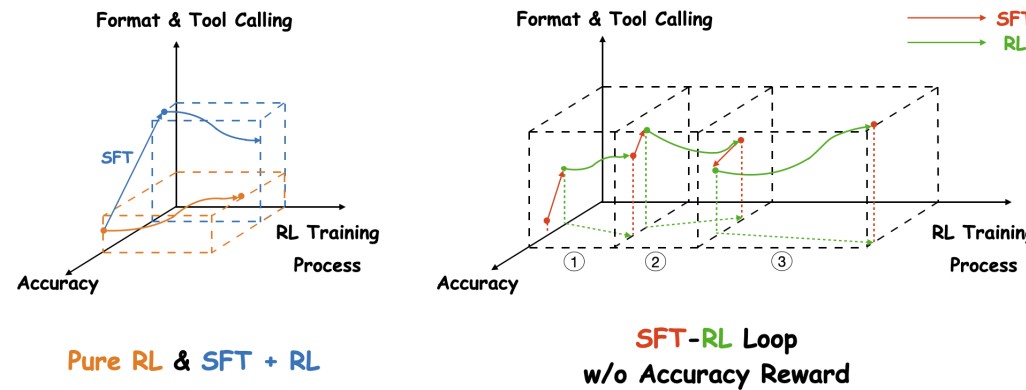

Figure 2: Conceptual illustration of our method. (**Left**) Pure RL fails to learn tool use; SFT + RL leads to suboptimal initialization. (**Right**) Epoch ①: SFT first instills partial tool-use capability. RL, operating *without accuracy-based rewards*, then explores and improves both tool use and accuracy, as shown by the green dashed projection of its trajectory onto the bottom surface. Epoch ②: RL without accuracy rewards explores suboptimal behaviors, which leads to a decline in model performance. Epoch ③: SFT corrects the model, guiding it back to the correct track and recovering accuracy. RL continues from this corrected state and further enhances the model's capability.

However, prior studies have shown that both approaches suffer from significant instability (Li et al., 2025a; Wu et al., 2025b; Li et al., 2025b). Figure 1 illustrates two central challenges in training LRMs for tool use: (i) **Pure RL** (top): Training with a tool-calling template and a reward signal that encourages tool invocation fails to elicit effective tool usage, resulting in a low tool-call ratio. (ii) **SFT + RL** (bottom): A cold-start pipeline first teaches tool usage through SFT, then refines it via RL. While this introduces the basic capability, the SFT initialization may be suboptimal, limiting RL effectiveness and making training vulnerable to early collapse. A further limitation of these methods is their dependence on ground-truth or verifiable accuracy signals. Yet, many real-world tasks—such as question answering (Li et al., 2025b), scientific report generation (Du et al., 2025), and summarization (Zhang et al., 2024)—lack such supervision. To mitigate this, some studies explore alternatives, including LLM-as-Judge (Gu et al., 2024; Zhou et al.) or training auxiliary reward models to approximate human preferences (Ouyang et al., 2022). However, these approaches face limited generalizability and impose substantial computational and resource costs.

In this work, we introduce a novel training framework, SFT-RL-Loop (`SR-Loop`), which leverages the complementary strengths of SFT and RL to train tool use in LRMs without requiring ground-truth supervision during RL—thereby reducing the need for supervised data. As shown in Figure 2, `SR-Loop` organizes training into iterative cycles with two stages: (1) SFT, which stabilizes learning by preserving output structure and constraining harmful exploration, and (2) RL, which encourages exploration and enables improvements beyond the fine-tuned policy. In the SFT stage, the model is trained on tool-calling data generated by a stronger off-policy model, allowing it to acquire proper tool-use templates. This stage also acts as a corrective mechanism, continually guiding the model back to the "right track" and preventing collapse. In the RL stage, we use Group Relative Policy Optimization (GRPO) (Shao et al., 2024) with rewards for correct output formatting and effective tool-use behavior, which does not need any ground truth supervision. This allows the model to autonomously explore and discover improved strategies while maintaining valid and structured outputs. By alternating between SFT and RL, `SR-Loop` maintains a balance between stability and exploration, enabling continuous progress without the need for explicit accuracy signals during RL.

We conduct comprehensive experiments to evaluate the performance of `SR-Loop` across a range of models and tasks, and observe consistent improvements in math reasoning when Python code is used as a reasoning tool. Although `SR-Loop` relies only on auxiliary reward signals such as output formatting, tool invocation behavior, and the balance between reasoning and tool usage, it achieves substantial accuracy gains over strong baselines. For instance, it improves Skywork OR1-Math 7B (He et al., 2025) by more than 3 points, and yields gains of 3.7 and 5.9 points on Qwen2.5-7B-SimpleRL and Qwen2.5-1.5B-SimpleRL (Zeng et al., 2025), respectively, all without using explicit accuracy supervision during RL training. These results demonstrate that the alternating SFT and RL

structure of `SR-Loop` enables stable and effective training for tool-augmented reasoning, even in the absence of ground-truth rewards during RL training. Furthermore, we evaluate `SR-Loop` in a tool free setting by applying it to text-only math reasoning tasks. Even without accuracy based rewards during RL and using only a format based reward signal, `SR-Loop` achieves strong performance on in domain math benchmarks and generalizes well to out of distribution reasoning tasks.

## 2 SFT-RL Loop

In this section, we introduce our `SR-Loop` approach, which iteratively alternates between SFT (Section 2.1) and RL (Section 2.2) phases to train LRMs for tool use without relying on ground-truth supervision during RL training (Section 2.3).

### 2.1 Cold-Start

We begin with SFT to help the model acquire tool-calling patterns. Given a dataset of complex tasks $\mathcal{X} = \{x_1, x_2, \cdots, x_n\}$, we leverage a stronger off-policy model, denoted as $\pi_{\text{off}}$, to generate tool-based solutions. To promote consistent tool usage, the generation is conditioned on a predefined `Tool Prompt`, which is specifically designed to trigger tool-calling behavior. This yields the SFT training dataset:

$$\mathcal{D}_{\text{SFT}} = \left\{ (x_i, y_i) \Big| y_i = \mathcal{LLM}_{\text{off}}(c_i, \text{Tool Prompt}), \ \forall x_i \in \mathcal{X} \right\}, \tag{1}$$

where $\mathcal{LLM}_{\text{off}}(x_i, \text{Tool Prompt})$ denotes the tool-augmented response generated by the off-policy model. Additionally, the dataset is verified for correctness through expert human review. Furthermore, we train our model $\pi_\theta$ on $\mathcal{D}_{\text{SFT}}$ using the standard SFT objective:

$$\mathcal{L}_{\text{cold start}} := -\mathbb{E}_{(x,y) \sim \mathcal{D}_{\text{SFT}}} \log \pi_\theta(y|x), \tag{2}$$

where $\pi_\theta(\cdot)$ denotes the reasoning model's likelihood function (or generation policy).

While the cold-start stage enables $\pi_\theta$ to acquire the syntactic ability to produce tool-based solutions by imitating demonstrations from $\pi_{\text{off}}$, it suffers from a key limitation: *distributional shift* (Ross et al., 2011; Quiñonero-Candela et al., 2022). Specifically, the model is trained on the static dataset $\mathcal{D}_{\text{SFT}}$, which consists of trajectories sampled from $\pi_{\text{off}}$, whereas at inference it generates outputs under its own policy. This discrepancy is especially problematic in sequential decision-making, where each output recursively shapes future inputs – even a small early error can push the model into unseen or low-probability states under $\pi_{\text{off}}$, increasing the likelihood of further mistakes. This results in *compounding errors*, with the total error scaling as $\mathcal{O}(T^2\epsilon)$ for trajectory length $T$ and per-step imitation loss $\epsilon$ (Ross et al., 2011). Since $\pi_\theta$ is never exposed to its own decision-induced states during SFT, it cannot recover from errors or adapt in off-distribution regions. Thus, while SFT offers a strong initialization for tool use, it does not guarantee robustness or generalization in long-horizon complex tasks.

### 2.2 Reinforcement Learning

To mitigate distributional shift, we incorporate RL to expose the model to its own trajectories, allowing it to learn corrective behaviors in off-distribution states by optimizing for task-level success rather than simply imitating expert demonstrations. To further encourage tool-calling, we design a tailored reward function $R(\cdot)$ that provides signals aligned with effective tool use. This reward function consists of three main components:

- **Format reward:** Encourages the model to structure its output according to predefined formatting rules. For instance, the final answer must be enclosed within `<answer>` and `</answer>`, while tool outputs (e.g., Python execution results) must be enclosed within `<interpreter>` and `</interpreter>`.
- **Tool use reward:** Provides feedback on the model's tool usage proficiency. The reward is positive when the model (i) attempts to invoke tools, (ii) executes tool calls successfully, and (iii) meaningfully incorporates tool outputs into its final response.

---

**Algorithm 1** `SR-Loop`: SFT-RL Iterative Training Framework

---

**Input:** Complex task dataset $\mathcal{X}$, off-policy model $\pi_{\text{off}}$, Tool Prompt, initial model $\pi_\theta$, reward function $R(\cdot)$, number of iterations $T$, RL training dataset $\mathcal{D}_{\text{RL}}$

**Output:** Trained model $\pi_\theta$

1: // Generate SFT data
2: $\mathcal{D}_{\text{SFT}} \leftarrow \left\{ \pi_{\text{off}}(x_i, \text{Tool Prompt}) \mid x_i \in \mathcal{X} \right\}$
3: // Partition datasets into T subsets
4: Split $\mathcal{D}_{\text{SFT}}$ into $\{\mathcal{D}_{\text{SFT}}^{(1)}, \cdots, \mathcal{D}_{\text{SFT}}^{(T)}\}$
5: Split $\mathcal{D}_{\text{RL}}$ into $\{\mathcal{D}_{\text{RL}}^{(1)}, \cdots, \mathcal{D}_{\text{RL}}^{(T)}\}$
6: **for** $i = 1$ to $T$ **do**
7:    // Step 1: Supervised Fine-Tuning
8:    $\pi_\theta \leftarrow \text{Train}(\pi_\theta, \mathcal{D}_{\text{SFT}}^{(i)}, \mathcal{L}_{\text{cold start}})$
9:    // Step 2: Reinforcement Learning (GRPO)
10:    **for** $q \in \mathcal{D}_{\text{RL}}^{(i)}$ **do**
11:       $\pi_{\theta_{\text{old}}} \leftarrow \pi_\theta$
12:       Sample $N$ trajectories $\{\tau_1, \cdots, \tau_N\} \sim \pi_{\theta_{\text{old}}}$
13:       Compute rewards $R(\tau_i)$ using format, tool-calling and balance criteria
14:       Compute normalized advantages $A_i$ and importance weights $r_{i,t}(\theta)$
15:       Update $\pi_\theta$ using GRPO loss $\mathcal{L}_{\text{GRPO}}$
16:    **end for**
17: **end for**

---

- **Reasoning-tool balance reward:** Promotes a balanced integration of reasoning and tool-calling behavior by countering the model's tendency to overuse tools without proper reasoning, as influenced by the previous two rewards.

Based on these components, the final reward is computed as:

$$\text{Reward} = 0.2 \times \text{Format Reward} + 0.5 \times \text{Tool Use Reward} + 0.3 \times \text{Balance Reward} \tag{3}$$

With the proposed reward design in Equation 3, we adopt GRPO (Shao et al., 2024) to train $\pi_\theta$ on the RL dataset $\mathcal{D}_{\text{RL}} = \{q_1, q_2, \cdots, q_n\}$ Specifically, for $q \in \mathcal{D}_{\text{RL}}$, we use the old policy from previous step $\pi_{\theta_{\text{old}}}$ to sample a group of $N$ individual responses $\tau_i$. Then, the RL loss is defined as:

$$\mathcal{L}_{\text{GRPO}}(\theta) = \mathbb{E}_{\tau_i \sim \pi_{\theta_{\text{old}}}(q), q \sim \mathcal{D}_{\text{RL}}} \frac{1}{\sum_{i=1}^{N} |\tau_i|} \sum_{i=1}^{N} \sum_{t=1}^{|\tau_i|} \text{CLIP}(r_{i,t}(\theta), A_i, \epsilon) - \beta \cdot \mathbb{D}_{\text{KL}}[\pi_\theta \| \pi_{\text{ref}}]. \tag{4}$$

where

$$A_i = \frac{R(\tau_i) - \text{mean}(\{R(\tau_i) \mid \tau_i \sim \pi_{\theta_{\text{old}}}(\tau), i = 1, 2, \ldots, N\})}{\text{std}(\{R(\tau_i) \mid \tau_i \sim \pi_{\theta_{\text{old}}}(\tau), i = 1, 2, \ldots, N\})}, \tag{5}$$

and $r_{i,t}(\theta) = \pi_\theta(\tau_{i,t}|q, \tau_{i,<t})/\pi_{\theta_{\text{old}}}(\tau_{i,t}|q, \tau_{i,<t})$. While RL mitigates distributional shift by training on trajectories sampled from the model's own policy, it introduces a new challenge: *entropy collapse* (Cui et al., 2025): As the model is optimized to maximize expected reward, the policy may collapse to a narrow set of high-reward actions, reducing exploration.

## 2.3 SFT-RL Loop (`SR-Loop`)

To effectively combine the complementary strengths of SFT and RL, we propose an iterative training framework called `SR-Loop`. This framework alternates between SFT and RL stages, progressively enhancing the model's tool-use capabilities while preserving structural consistency and training stability. To support iterative learning, we partition both the SFT and RL datasets, $\mathcal{D}_{\text{SFT}}$ and $\mathcal{D}_{\text{RL}}$, into $T$ subsets, denoted as $\{\mathcal{D}_{\text{SFT}}^{(1)}, \cdots, \mathcal{D}_{\text{SFT}}^{(T)}\}$ and $\{\mathcal{D}_{\text{RL}}^{(1)}, \cdots, \mathcal{D}_{\text{RL}}^{(T)}\}$, respectively. Here, $T$ is a hyperparameter that specifies the total number of training iterations in the `SR-Loop` cycle.

At each iteration $i$, the model first undergoes SFT using the loss defined in Equation 2 on $\mathcal{D}_{\text{SFT}}^{(i)}$. This step reinforces correct tool-calling patterns and stabilizes model behavior, particularly by restoring

structured outputs after potential deviations during RL exploration. Subsequently, the model enters the RL stage, where it is optimized on dataset $\mathcal{D}_{\text{RL}}^{(i)}$ using the GRPO objective defined in Equation 4, guided by the reward function $R(\cdot)$ introduced earlier. This stage encourages exploration beyond the expert's distribution, allowing the model to discover more effective tool-use strategies and improve task-level performance.

By alternating between these two stages, `SR-Loop` enables the model to iteratively refine its reasoning and tool-use behaviors. In the SFT stage, the model not only learns tool-calling syntax and formatting from off-policy demonstrations but also benefits from periodic correction, which mitigates drift caused by RL and realigns the model with high-quality exemplars. In the RL stage, the model is trained on its own sampled trajectories using weak yet informative rewards focused on output structure, tool-use proficiency, and reasoning-tool balance. Importantly, **no explicit accuracy reward is required**; instead, the model learns to improve task performance implicitly through exploration and structured feedback. An overview of the complete training loop is provided in Algorithm 1. We propose a detailed analysis about convergence properties of `SR-Loop` in Appendix A.1.

# 3 EXPERIMENT

## 3.1 SETUP

**Training Datasets:** We primarily utilize the Retool-SFT dataset (Feng et al., 2025) during the SFT stage, which is generated using the tool-use template by Deepseek-R1 (Guo et al., 2025) and double-checked by human experts. For the RL stage, we employ the DeepScaleR dataset (Luo et al., 2025), which consists of 40,000 mathematics problems compiled from AIME and AMC competitions held before 2023, as well as from Omni-MATH (Gao et al., 2024).

**Tools:** In the case of solving mathematical problems, the model leverages tools for code generation and execution. Specifically, it generates Python code, executes it using an external interpreter, and incorporates the returned result into its subsequent reasoning process.

**Benchmarks:** To assess mathematical reasoning proficiency, we employ four benchmarks: MATH-500 (Lightman et al., 2023), a 500-question subset of the MATH benchmark curated for rigorous; Minerva Math (Hendrycks et al., 2021); AIME24; Olympiad Bench (He et al., 2024) as well as competition-level benchmarks such as AMC23 (AMC, 2023). We evaluate the performance of the model on each dataset using the `pass@1` measure.

**Baselines:** We compare our method against state-of-the-art tool-use training baselines, all trained on the same dataset for a fair comparison: (i) **Original** refers to a baseline model trained without any tool-use capability, serving as a reference point for improvements from tool integration; (ii) **SFT + RL** represents the standard two-stage approach that first applies SFT, followed by RL for further optimization; (iii) **SFT loss** augments the RL objective with an additional SFT loss on positive examples generated by the model itself during rollouts, using a combined loss of $\mathcal{L}_\theta = \mathcal{L}_{\text{RL}}(\theta) + \mathcal{L}_{\text{SFT}}^{\text{self-rollout}}(\theta)$; (iv) **LUFFY** (Yan et al., 2025) introduces a dynamic weighting mechanism that balances imitation and exploration by integrating off-policy demonstrations with on-policy rollouts throughout training; and (v) **SR-Loop with accuracy reward** (`SRL (w/ acc)`) implements our proposed `SR-Loop` strategy but includes an explicit accuracy-based reward during the RL phase, contrasting with our main setting which relies solely on structural and behavioral feedback.

**Models:** We evaluate three widely-used open-source LLMs as backbone models, spanning different architectures and model sizes. Skywork-OR1-Math-7B (He et al., 2025) is derived from DeepSeek-R1-Distill-Qwen-7B (Guo et al., 2025) through RL and is specifically optimized for mathematical reasoning. Qwen-2.5-7B-SimpleRL and Qwen-2.5-1.5B-SimpleRL (Zeng et al., 2025) are based on Qwen-2.5 (Yang et al., 2024), further refined through a lightweight RL process.

**Implement Details:** For the SFT stage, we employed the LLaMA-Factory library (Zheng et al., 2024), a widely adopted GitHub-hosted framework for efficient large-model fine-tuning, to carry out all training procedures. Experiments are conduced on four 141GB NVIDIA H200 GPUs, with learning rate as $1 \times 10^{-5}$, global batch size as 32, and max token as 8192. Furthermore, to reduce memory consumption during training, we applied ZeRO Stage-2 optimization and gradient checkpointing, both provided by the DeepSpeed library. For RL training, we use the EasyR1 (Zheng et al., 2025) framework built on verl (Sheng et al., 2024), with specialized support for VLMs. Experiments

Table 1: Main results of `SR-Loop` and `SRL` (w/ acc). "Tool" indicates whether the model acquires tool-use capabilities. The "Average" column reports the mean accuracy across all five math reasoning benchmarks. Relative improvements over the Original model are shown in superscript.

| Model | Method | Tool | Math Reasoning | | | | | Average |
|---|---|---|---|---|---|---|---|---|
| | | | MATH-500 | AMC | AIME | Olympiad | Minerva | |
| **Skywork-OR1 -Math-7B** | Original | ✗ | 86.4 | 74.5 | 53.3 | 55.3 | 36.7 | 61.2 |
| | SFT + RL | ✓ | 84.6 | 75.9 | 56.7 | 57.4 | 37.1 | $62.3^{+1.1}$ |
| | SFT loss | ✗ | 87.2 | 72.3 | 53.3 | 54.1 | 36.0 | $60.6^{-0.6}$ |
| | LUFFY | ✗ | 83.0 | 73.5 | 56.7 | 56.5 | 37.1 | $61.4^{+0.2}$ |
| | SRL (w/ acc) | ✓ | 90.4 | 78.2 | 63.3 | 60.2 | 37.9 | $66.0^{+4.8}$ |
| | SR-Loop | ✓ | 88.8 | 76.8 | 60.0 | 59.6 | 37.8 | $64.6^{+3.4}$ |
| **Qwen-2.5-7B -SimpleRL** | Original | ✗ | 73.2 | 45.8 | 16.7 | 38.7 | 25.7 | 40.0 |
| | SFT + RL | ✓ | 78.6 | 47.0 | 20.0 | 40.8 | 25.9 | $42.5^{+2.5}$ |
| | SFT loss | ✗ | 73.8 | 45.8 | 16.7 | 40.2 | 25.4 | $40.4^{+0.4}$ |
| | LUFFY | ✗ | 74.4 | 44.6 | 23.3 | 39.9 | 26.8 | $41.8^{+1.8}$ |
| | SRL (w/ acc) | ✓ | 79.8 | 50.4 | 23.3 | 44.1 | 27.6 | $45.0^{+5.0}$ |
| | SR-Loop | ✓ | 79.4 | 49.3 | 20.0 | 42.5 | 27.3 | $43.7^{+3.7}$ |
| **Qwen-2.5-1.5B -SimpleRL** | Original | ✗ | 21.0 | 16.9 | 3.3 | 11.1 | 11.4 | 12.7 |
| | SFT + RL | ✓ | 26.2 | 18.5 | 3.3 | 12.4 | 13.1 | $14.7^{+2.0}$ |
| | SFT loss | ✗ | 18.8 | 15.7 | 6.7 | 10.3 | 10.9 | $12.5^{-0.2}$ |
| | LUFFY | ✗ | 29.6 | 20.5 | 6.7 | 12.9 | 13.3 | $16.6^{+3.9}$ |
| | SRL (w/ acc) | ✓ | 34.6 | 24.1 | 13.3 | 16.3 | 17.2 | $21.1^{+8.4}$ |
| | SR-Loop | ✓ | 31.4 | 21.7 | 10.0 | 14.3 | 15.7 | $18.6^{+5.9}$ |

are conducted using eight 141GB NVIDIA H200 GPUs with a global batch size of 48, a rollout batch size of 16, a rollout temperature of 1.0, a consistent learning rate of $1 \times 10^{-6}$, and 8 rollouts. Additionally, to mitigate the risk of model overfitting caused by repeated exposure to the same corpus, we uniformly partitioned the dataset into multiple non-overlapping subsets of equal size.

## 3.2 Main Results

**`SR-Loop` improves tool use and accuracy without accuracy-based rewards.** As shown in Table 1, our `SR-Loop` method consistently enhances model performance even without access to any accuracy-based reward during RL training. On Skywork-Math-7B, `SR-Loop` achieves an average score of 64.6, outperforming both LUFFY (61.4), SFT loss (60.6), and the full SFT+RL pipeline (62.3). Similar improvements are observed for Qwen2.5-7B-SimpleRL (43.7 vs. 41.8 for LUFFY, 40.4 for SFT loss and 42.5 for SFL+RL), and on the smaller Qwen2.5-1.5B-SimpleRL model (18.6 vs. 16.6, 12.5 and 14.7 respectively). Notably, `SR-Loop` also learns to invoke tools correctly, without any explicit reward signal for accuracy, indicating that the SFT and RL loop structure itself provides a strong training signal that keeps the model on a meaningful learning trajectory. This suggests that looped optimization alone can help stabilize training even under weak supervision.

**Adding accuracy rewards further boosts performance.** When we incorporate programmatic accuracy signals back into the RL stage, as in `SRL` (w/ acc), we see further notable gains. On Skywork-Math-7B, accuracy rises to 66.0, the highest among all methods. For Qwen2.5-7B-SimpleRL, `SRL` (w/ acc) reaches 45.0, significantly outperforming all baselines. Even for the small Qwen2.5-1.5B-SimpleRL, accuracy climbs to 21.1, showing a remarkable +8.4 absolute gain over the original model. These results confirm that adding verifiable accuracy signals complements the SFT-RL loop: the loop provides structure and stability and helps the model escape local minima and training collapse. Together, they enable robust and scalable reasoning across model sizes and tasks.

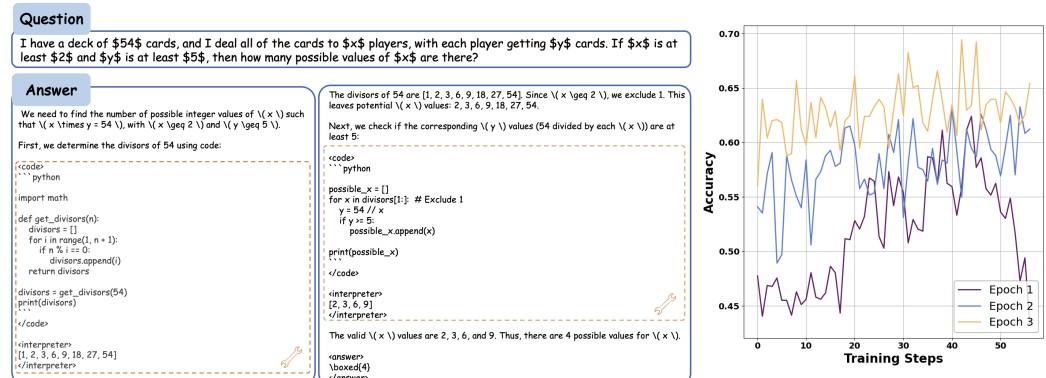

Figure 3: Concrete Example.

Figure 4: Training Process.

## 3.3 ANALYSIS

**Concrete Examples** Figure 3 shows a representative output from a model trained with `SR-Loop`. The model Skywork-OR1-Math-7B presents a clear natural language explanation followed by well-structured, executable code. It correctly calls tools, formats its output with formal expressions and boxed answers, and organizes the reasoning into modular steps that mirror the problem structure. This output reflects both the formatting conventions reinforced during SFT and the reasoning flexibility encouraged by RL.

**Training Process** Figure 4 illustrates the model Skywork-OR1-Math-7B's performance trajectory across three full `SR-Loop` epochs, showing steady improvements in accuracy over training steps. Each epoch reflects one complete cycle of alternating SFT and RL phases over the corresponding partitioned datasets. Notably, accuracy consistently improves from Epoch 1 to Epoch 3, demonstrating the effectiveness of our iterative training strategy. In Epoch 1 (purple), the model exhibits unstable performance, likely due to early-stage exploration during RL that introduces output variability. However, in Epoch 2 (blue), we observe a marked increase in accuracy and reduced variance, indicating that the SFT stage has successfully realigned the model with structured, high-quality behavior while retaining beneficial exploration gains. By Epoch 3 (orange), the model achieves both higher average accuracy and more stable performance, showing that the SR-Loop's alternation between SFT correction and RL-driven exploration leads to cumulative improvements. This progressive refinement suggests that the model gradually learns to balance structured tool use and effective reasoning, ultimately achieving better task-level performance without relying on explicit accuracy-based rewards. The visualization affirms that our method enables stable and effective learning dynamics over time.

**Correlation of Accuracy Reward and Other Rewards** To assess how well non-accuracy rewards align with model accuracy, we sample several checkpoints during the training process of `SR-Loop`, both with and without the accuracy reward. Specifically, we compute the correlation between accuracy and other reward components, including format, tool-calling, and balance rewards. As shown in Figure 5, the model trained with the accuracy reward (blue) exhibits a slightly higher correlation (0.5695) compared to the model trained without it (yellow, 0.4554). Nevertheless, the strong correlation in the absence of the accuracy signal indicates that our SFT+RL loop method effectively keeps the model on the correct track. These results demonstrate that carefully designed non-accuracy rewards are sufficient to guide the model toward accurate behavior, eliminating the need for explicit accuracy supervision during RL training.

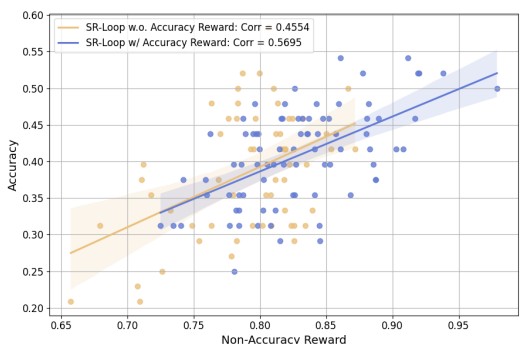

Figure 5: Correlation of accuracy and other rewards during RL training, with (blue) and without (yellow) the accuracy reward.

Table 2: Performance of `SR-Loop` in training general reasoning models without tool use.

| Method | Math Reasoning | | | | | | Out-of-Distribution | | | |
|--------|----------|-----|------|----------|---------|------|-------|--------|---------|------|
| | MATH-500 | AMC | AIME | Olympiad | Minerva | Avg. | ARC-C | GPQA-D | MMLU-Pro | Avg. |
| Original | 48.8 | 32.6 | 11.4 | 15.8 | 8.7 | 23.5 | 18.2 | 11.1 | 16.9 | 15.4 |
| SFT + RL | 84.2 | 67.1 | 32.5 | 54.6 | 34.1 | 54.5 | 76.4 | 37.9 | 49.6 | 54.6 |
| SimpleRL | 76.0 | 54.9 | 27.0 | 34.7 | 25.0 | 43.5 | 30.2 | 23.2 | 34.5 | 29.3 |
| LUFFY | 87.6 | 65.6 | 29.4 | 57.2 | 37.5 | 55.5 | 80.5 | 39.9 | 53.0 | 57.8 |
| TAPO | 83.4 | 77.5 | 33.3 | 46.2 | 38.2 | 55.7 | 81.6 | 37.9 | 49.6 | 56.4 |
| SR-Loop | 85.6 | 69.6 | 35.3 | 57.4 | 37.1 | $57.1^{+1.4}$ | 83.4 | 40.9 | 55.9 | $60.1^{+2.3}$ |

## 4 GENERAL TRAINING WITHOUT TOOL

In a more general setting, where the model is trained to improve its reasoning capabilities, we implement the `SR-Loop` to demonstrate its generalizability.

### 4.1 SETUP

**Training Datasets** In the SFT stage, we utilize the OpenR1-Math-46k-8192 dataset[1] (Yan et al., 2025), a 46,000-example subset of OpenR1-Math-220k (Face, 2025), containing mathematical problems drawn from NuminaMath 1.5 (LI et al., 2024) and paired with high-quality reasoning responses generated by DeepSeek-R1 (Guo et al., 2025). For the RL stage, we use the DeepScaleR dataset (Luo et al., 2025), which comprises 40,000 math problems sourced from pre-2023 AIME and AMC competitions as well as Omni-MATH (Gao et al., 2024).

**Baselines** In the context of general training, we evaluate several state-of-the-art algorithms that combine SFT and RL in different ways. Specifically, we include: (i) **SimpleRL-Zoo** (Zeng et al., 2025), which applies GRPO to approximately 24k mathematical samples from GSM8K and MATH; (ii) **SFT + RL**, the standard two-stage approach that first applies SFT, followed by RL for further optimization; (iii) **LUFFY** (Yan et al., 2025), which introduces a dynamic weighting mechanism to balance imitation and exploration by integrating off-policy demonstrations with on-policy rollouts during training; (iv) **TAPO** (Wu et al., 2025a), which dynamically incorporates structured external knowledge within the GRPO framework.

**Evaluation** We evaluate the model on five mathematical reasoning benchmarks: AIME24, AMC, Minerva, OlympiadBench, and MATH500, as described in Section 3. For datasets with limited sample sizes, specifically AIME24 and AMC, we report the avg@32 metric, while for the remaining benchmarks we adopt pass@1 as the evaluation criterion. To further assess the model's generalization ability beyond mathematical reasoning, we also evaluate it on three out-of-distribution benchmarks: ARC-C (Clark et al., 2018), which focuses on open-domain reasoning; GPQA-Diamond (Rein et al., 2024), which targets graduate-level scientific knowledge and is denoted as GPQA-D; and MMLU-Pro (Wang et al., 2024c), which includes reasoning problems drawn from academic examinations and textbooks.

**Reward** We simply apply a format reward to guide general reasoning models, such as placing the reasoning process within `<think>...</think>` tags and the final answer within `<answer>...</answer>` tags.

### 4.2 MAIN RESULT

We conduct experiments on Qwen2.5-Math-7B (Yang et al., 2024). As shown in Table 2, `SR-Loop` delivers consistently strong performance across five mathematical reasoning benchmarks, achieving the highest average accuracy of 57.1 and outperforming leading baselines such as LUFFY (55.5) and TAPO (55.7). Notably, even without using an explicit accuracy reward, `SR-Loop` improves

---

[1]`https://huggingface.co/datasets/Elliott/Openr1-Math-46k-8192`

upon conventional SFT+RL pipelines and other advanced methods, demonstrating the effectiveness of its self-refinement mechanism. Beyond in-domain math tasks, `SR-Loop` also exhibits excellent generalization to out-of-distribution benchmarks—including ARC-C, GPQA-Diamond, and MMLU-Pro—achieving the best average score of 60.1, a +2.3 gain over the strongest baseline. These results highlight the robustness and broad transferability of `SR-Loop` across diverse reasoning tasks, even in the absence of external tools or structured supervision.

## 5 RELATED WORK

**Tool Use in LRMs**   LRMs (Guo et al., 2025; Yang et al., 2025; Seed et al., 2025; Team et al., 2025) excel at complex reasoning (Wang et al., 2024a; Hsiao et al., 2025; Shi et al., 2025; Qu et al., 2025) but are limited by static pretraining knowledge (Gao et al., 2023; Zhu et al., 2025; Wang et al., 2024b; Cheng et al., 2024; Matarazzo & Torlone, 2025). To overcome this, recent work augments LRMs with external tools (Qu et al., 2025; Mei et al., 2025), enabling dynamic interaction with sources like Python interpreters (Forootani, 2025), web search (Jin et al., 2025), and worker LLMs (Guo et al., 2024; Team et al., 2025; Li et al., 2025b; Feng et al., 2025; Gim et al., 2024). This reduces hallucination and enhances symbolic reasoning (An et al., 2025; Niketan et al., 2024), allowing models to produce grounded outputs and perform deterministic tasks (Forootani, 2025). Tool integration methods are typically tuning-free (e.g., ICL prompts (Qu et al., 2024)) or tuning-based (e.g., SFT on tool-use data (Zhang et al., 2025b; Liu et al., 2024)). Toolformer introduced self-supervised API call learning (Schick et al., 2023), while later work uses RL with execution-based rewards (Zhang et al., 2025a). The toolbox now includes databases, expert models, and physical systems (An et al., 2025), enabling LRMs to act as autonomous agents that decompose and execute multi-step plans (Qu et al., 2025).

**SFT and RL Training of LRMs**   Prior work shows that both pure RL and SFT-then-RL approaches often suffer from instability (Li et al., 2025a; Wu et al., 2025b; Li et al., 2025b). In pure RL, models trained to use tools via reward signals frequently fail to learn effective tool use, resulting in low tool-call rates (Qian et al., 2025; Chu et al., 2025). The field has since shifted toward RL for Verifiable Reasoning (RLVR), which uses objective rewards on tasks with ground truth, like math and code (Zhang et al., 2025a). A more common approach combines SFT and RL—first training on tool use via imitation, then refining with RL (Chen et al., 2025a). However, SFT can overfit to static data and generalize poorly, especially in multi-step or unfamiliar settings (Chu et al., 2025; Qian et al., 2025). This rigidity can destabilize subsequent RL (Chen et al., 2025a). Moreover, both paradigms rely on verifiable rewards, which are scarce in open-ended tasks like QA (Li et al., 2025b), scientific writing (Du et al., 2025), or summarization (Zhang et al., 2024). To address this, recent work explores using LLMs as judges (Gu et al., 2024; Zhou et al.) or training reward models from human preferences (Ouyang et al., 2022). Yet these face issues like bias, variance, and annotation costs (Zhang et al., 2025a). Emerging efforts aim to learn from both successes and failures via preference data from tool-use trajectories, applying optimization methods like DPO (Chen et al., 2025b; Jung et al., 2025).

## 6 CONCLUSION

We propose `SR-Loop`, a general training framework that alternates between SFT and RL to improve tool-augmented reasoning without relying on accuracy-based rewards during RL. By leveraging weak but structured signals—such as format, tool-use, and balance rewards—`SR-Loop` enables stable training, avoids entropy collapse, and supports exploration beyond static SFT data. Experiments across multiple model sizes and benchmarks show that `SR-Loop` consistently improves both tool-use and task accuracy, outperforming strong baselines. When accuracy rewards are reintroduced, performance improves further, validating the synergy between SFT and RL. Beyond tool use, `SR-Loop` also generalizes to tool-free reasoning tasks, achieving strong in-domain and out-of-distribution performance using only format rewards. These results demonstrate that alternating between SFT and RL provides a robust and scalable solution for training reasoning models in settings where ground-truth supervision is limited or unavailable.

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

## LLM USAGE

We used LLMs as general-purpose writing and debugging assistants. Specifically, LLMs were employed to help polish the writing (e.g., improving sentence clarity, grammar, and flow) and occasionally to assist with debugging minor implementation issues (e.g., identifying syntax errors or suggesting code refactoring). However, all core ideas, research questions, methodological designs, codebase implementations, experiments, and analyses were entirely conceived, developed, and conducted by the authors. No part of the intellectual contribution, experimental framework, or scientific reasoning was generated by an LLM.

## LIMITATION

While `SR-Loop` demonstrates stable and scalable learning without relying on accuracy-based rewards, it has several limitations. First, the approach still requires access to a capable off-policy model to generate high-quality, tool-augmented responses during the SFT phase. These off-policy demonstrations play a critical role in bootstrapping tool-use behavior and guiding the model's initial learning. However, in domains where such expert models are unavailable, underperforming, or prohibitively expensive to fine-tune or generate data from, applying `SR-Loop` may become impractical or less effective. This reliance introduces a bottleneck that may hinder adoption in low-resource or rapidly evolving domains. Second, its scalability to more complex, multi-turn, or interactive settings remains an open question. In such environments—such as web agents, dialogue systems, or real-time information retrieval—the model must reason over non-deterministic tool outputs, partial observability, and dynamically evolving contexts. It is unclear whether the current SFT-RL alternating structure is sufficient to handle these challenges, or whether additional mechanisms (e.g., memory, planning, or hierarchical control) are needed to support stable learning and generalization in these more demanding scenarios.

## A  APPENDIX

### A.1  CONVERGENCE PROCESS OF SR-LOOP

`SR-Loop` alternates between RL and SFT to improve a model's tool-use and reasoning capabilities without relying on accuracy-based supervision. In the RL phase, the model policy $\pi_\theta$ is updated to maximize a proxy reward signal $r$, which captures desirable behaviors such as correct output formatting, appropriate tool usage, or balanced reasoning strategies. These proxy rewards are dense and practical to compute, but only partially aligned with true task success. Optimizing them directly can introduce drift or brittle behaviors that exploit the reward signal without improving actual performance. To correct for this, each RL phase is followed by an SFT phase, where the model is fine-tuned on high-quality responses generated by an off-policy teacher $\pi_{\text{off}}$. This step minimizes a supervised loss, which acts as a projection that pulls the current policy back toward expert-like behavior. The SFT step repairs structural issues, recovers entropy, and prevents collapse into narrow or reward-hacking solutions.

Over successive iterations, this alternating procedure leads to convergence. The RL phase explores new behaviors and improves proxy reward, while the SFT phase regularizes that exploration by reinforcing well-structured, interpretable outputs. Even though the reward signal $r$ is not directly tied to ground-truth correctness, the combination of exploration and correction steadily improves the model's ability to reason and use tools effectively. The result is a stable policy that balances learned reward optimization with consistency to expert-like behavior, enabling long-term improvement without requiring access to explicit accuracy labels during RL training.

