# OpenReview forum: "Enabling Tool Use of Reasoning Models Without Verifiable Reward via SFT-RL Loop"
_ICLR.cc/2026/Conference — Submitted to ICLR 2026_

### Official Review · Reviewer_cUV8 · 2025-10-30

**Soundness:** 2
**Presentation:** 3
**Contribution:** 3
**Rating:** 4
**Confidence:** 4

**Summary:**

This paper addresses the severe instability that occurs when training large reasoning models to use tools, particularly during reinforcement learning (RL). The authors propose an alternating training approach between supervised SFT and RL. SFT stabilizes learning, preserves output structure, and constrains harmful exploration. RL encourages exploration while ensuring improvement over the fine-tuned policy. During RL, the authors train the models' tool use capabilities using Format Reward and Tool Use Reward rather than ground-truth Accuracy reward. They believe this allows the model to autonomously explore and discover improved strategies while maintaining valid and structured outputs. Their experimental results show that relying solely on auxiliary reward signals during RL can bring substantial accuracy improvements.

**Strengths:**

The authors attempt to combine SFT and RL, leveraging the strengths of both to address the instability problem that exists in tool RL training, which is an exploration of a new approach.

**Weaknesses:**

1. In Section 3, the baselines used for comparison do not seem entirely fair — most of them do not use tools, such as LUFFY. Moreover, the paper does not compare against other works related to Tool-Integrated Reasoning, such as SimpleTIR [1] or ZeroTIR [2].

2. From the experimental results (e.g., Table 1), it appears that adding an accuracy reward constraint yields more stable results than omitting it. Therefore, the authors’ advocacy for training without ground-truth accuracy rewards does not seem strongly justified.

3. Although the authors treat SFT and RL as one complete cycle of alternating SFT and RL phases within an epoch, it is unclear how much each contributes to the overall accuracy. For instance, in Figure 4, if both phases take up half of the training steps, the RL phase seems to contribute very little to accuracy improvement.

4. Again, from Figure 4, we can see that SR-Loop has relatively few training steps overall, and accuracy gradually converges toward the end. This makes it difficult to determine whether, in long-term training, omitting supervision from ground-truth accuracy rewards might lead to “hacking” behavior.

[1] SimpleTIR: End-to-End Reinforcement Learning for Multi-Turn Tool-Integrated Reasoning
[2] Agent RL Scaling Law: Spontaneous Code Execution for Mathematical Problem Solving

**Questions:**

1. Could the authors include additional Tool-Integrated Reasoning–related baselines in the experiments, such as SimpleTIR?

2. It would be helpful to see how SFT and RL each contribute to the model’s performance. This could support the authors’ claim that “SFT serves to stabilize learning, preserve output structure, and constrain harmful exploration, while RL encourages exploration that improves upon the fine-tuned policy.”

3. Could the authors observe the changes in reward and accuracy of SR-Loop training over a longer training period?

---

### Official Review · Reviewer_ynSk · 2025-10-31

**Soundness:** 1
**Presentation:** 2
**Contribution:** 2
**Rating:** 4
**Confidence:** 4

**Summary:**

The paper proposes SR-Loop, an iterative SFT→RL training scheme for tool-augmented reasoning that do not need to use accuracy-based rewards during RL.

**Strengths:**

1. Achieves competitive results without explicit accuracy rewards.

**Weaknesses:**

1. The method extends two-stage SFT+RL with multi iteration training, but do not discuss about the training stability on the loop schedule.
2. The only tool evaluated is code, breadth beyond code execution is unclear.
3. The reward design is under-specified, details for terms such as the reasoning–tool balance are missing.

**Questions:**

1. How are the tool-use reward and the reasoning–tool balance computed?
2. Does the loop expose the policy to more on-policy data, and how do you separate method gains from extra data?
3. What is the rationale for the chosen reward weights, and can you provide ablations to verify each reward’s effectiveness and show which one contribute most?
4. With training/eval confined to a code sandbox, do the results generalize to multi-tool scenarios?
5. Adding an explicit accuracy reward (SRL w/ acc) improves performance, where does it help most, and at what weight?

---

### Official Review · Reviewer_13nM · 2025-10-31

**Soundness:** 3
**Presentation:** 2
**Contribution:** 2
**Rating:** 4
**Confidence:** 3

**Summary:**

- Proposes SR‑Loop, an alternating Supervised Fine‑Tuning (SFT) and Reinforcement Learning (RL) framework to train tool use without accuracy-based verifiable rewards during RL. The RL uses structured proxy rewards (format, tool invocation success, and reasoning–tool balance), while SFT periodically re‑aligns outputs to expert demonstrations to avoid collapse and drift

- Demonstrates improvements on math tool‑use tasks (Python execution) across Skywork‑OR1‑Math‑7B and Qwen2.5 variants, showing higher pass@1 than SFT+RL and other baselines; adding an accuracy reward on top of SR‑Loop further boosts performance

- Extends to tool‑free reasoning: using only format rewards, SR‑Loop improves in‑domain math and generalizes to out‑of‑distribution benchmarks (ARC‑C, GPQA‑Diamond, MMLU‑Pro)

**Strengths:**

Quality
- Comprehensive math evaluation across multiple models/benchmarks; includes ablations (SFT‑only vs. SFT→GRPO, SR‑Loop with/without accuracy reward) and reward–accuracy correlation analysis, supporting claims of stability and gains

Clarity
- The training loop (Algorithm 1), reward composition, and dataset setup are explained clearly; figures illustrate failure modes of pure RL and the corrective role of SFT in the loop

Significance
- Addresses common instability of tool‑use RL (entropy collapse, drift) and scarce verifiable rewards in real tasks; results suggest a robust recipe applicable beyond tool use

**Weaknesses:**

Dependence on off‑policy expert data
- SR‑Loop requires a capable off‑policy model and human‑verified SFT traces; feasibility in domains without high‑quality demonstrations remains uncertain (noted in Limitations)

Proxy reward fidelity and gaming
- Format/tool/balance rewards are weakly aligned with task success; more evidence is needed on robustness to reward hacking and failure cases, especially in multi‑turn settings

Evaluation scope
- Focused on math and Python tool execution; lacks results for heterogeneous, non‑deterministic tools (web search, DBs), multi‑turn agents, or mixed simulated/live environments

Novelty

- Alternating SFT and RL with periodic re-alignment to expert traces while using non‑accuracy proxy rewards (format, tool‑use, balance) is closely related to prior alternating or cooperative SFT↔RL recipes, off‑policy–guided rollouts, and DAgger‑style loops, as well as recent tool‑use RL that relies on structural/execution rewards rather than verifiable accuracy. Unless the authors more sharply differentiate SR‑Loop (e.g., with a distinct learning principle, theoretical guarantees, or convergence properties) and run stronger, controlled head‑to‑head comparisons against cooperative SFT+RL, DAgger‑like alternation, and tool‑reward baselines, the contribution risks being seen as incremental packaging of known ingredients rather than a fundamentally new algorithm. Concretely, the paper would benefit from ablations that (i) hold total SFT budget constant while varying loop cadence, (ii) swap in alternative proxy‑reward shapes to show the loop—not the reward—drives gains, and (iii) extend beyond math/Python to heterogeneous, non‑deterministic tools to demonstrate broader novelty and impact

**Questions:**

See above

---

### Official Review · Reviewer_XKve · 2025-11-01

**Soundness:** 2
**Presentation:** 3
**Contribution:** 2
**Rating:** 4
**Confidence:** 4

**Summary:**

This paper investigates the training of large reasoning models with tool utilization, and proposes a novel framework named SR-Loop, which is equipped with two alternating phases. The first one is SFT phase, in which the model is trained using SFT data to acquire tool-use capabilities. The second one is RL phase, in which the model is encouraged to explore improved strategies. Several experimental studies are conducted to verified its effectiveness.

**Strengths:**

- The paper is logically well structured and easy to follow.
- The authors provide detailed experimental analyses demonstrating the superiority of **SR-Loop**.

**Weaknesses:**

- In **Line 143**, it is unclear how the computational complexity $O(T^2 \epsilon)$ is derived. The authors are encouraged to provide a high-level explanation of this result.
- It remains unclear why a **distributional shift** problem arises after the SFT stage. The reviewer suggests adding an experiment to verify this claim.
- The computation of the **Reasoning–Tool Balance Reward** is not clearly described.
- In **(3)**, how to determine the weight of each term is not explained. An ablation study on the weight settings is necessary.
- The paper includes too few baselines and lacks comparisons with tool-related baselines such as **Search-R1** [1].

---
[1] Search-R1: Training llms to reason and leverage search engines with reinforcement learning. 2025.

**Questions:**

See them in Weaknesses.

---

### Meta-Review · Area_Chair_DKVG · 2026-01-04

**Summary:**

SR-Loop proposes an iterative SFT-RL training loop for tool-using reasoning models in settings where verifiable/accuracy rewards are unavailable during RL. The method alternates between (i) SFT on high-quality tool-use traces to maintain structure and correct drift, and (ii) GRPO-style RL optimizing proxy rewards such as format correctness, tool execution success, and a reasoning–tool balance term; experiments mainly on math with Python tools show gains over several SFT+RL baselines, and an additional setting reports improvements with format-only rewards.

Reviewers’ concerns are consistently negative and center on: (1) insufficient specification and ablations of the proxy rewards, especially the balance reward and weight sensitivity, raising reproducibility/robustness issues; (2) limited evaluation scope (largely Python-for-math), weakening claims of general tool-use; (3) incomplete or potentially unfair baselines, with missing comparisons to several relevant tool-integrated reasoning methods; (4) questioned novelty relative to prior alternating SFT/RL or DAgger-like loops; and (5) motivation mismatch, since results suggest adding accuracy rewards can help, and the contribution of RL vs repeated SFT/on-policy data is not cleanly disentangled, with limited long-horizon analysis for reward hacking/stability.

**Reviewer Concerns:**

Since the authors did not provide a rebuttal, none of the concerns were addressed.

**Reviewer Scores:**

Since the authors did not provide a rebuttal, the reviewers’ scores would not have increased.

---

### Decision · Program_Chairs · 2026-01-26

Reject